# Outcomes of isoniazid preventive therapy among people living with HIV in Kenya: A retrospective study of routine health care data

Muthoni Karanja[1]*, Leonard Kingwara[1,2], Philip Owiti[3,4], Elvis Kirui[2], Faith Ngari[1], Richard Kiplimo[3], Maurice Maina[4], Enos Masini[5], Elizabeth Onyango[3], Catherine Ngugi[1]

1 National AIDS and STI Control Program (NASCOP), Ministry of Health, Nairobi, Kenya, 2 National Public Health Laboratories, Ministry of Health, Nairobi, Kenya, 3 National Tuberculosis, Leprosy and Lung Disease Program (NTLD-P), Ministry of Health, Nairobi, Kenya, 4 United States Agency for International Development (USAID), Nairobi, Kenya, 5 Global Fund, Geneva, Switzerland

* evemuthoni2010@gmail.com

## Abstract

### Introduction

Isoniazid preventive therapy (IPT) taken by People Living with HIV (PLHIV) protects against active tuberculosis (TB). Despite its recommendation, data is scarce on the uptake of IPT among PLHIV and factors associated with treatment outcomes. We aimed at determining the proportion of PLHIV initiated on IPT, assessed TB screening practices during and after IPT and IPT treatment outcomes.

### Methods

A retrospective cohort study of a representative sample of PLHIV initiated on IPT between July 2015 and June 2018 in Kenya. For PLHIV initiated on IPT during the study period, we abstracted patient IPT uptake data from the National data warehouse. In contrast, we obtained information on socio-demographic, TB screening practices, IPT initiation, follow up, and outcomes from health facilities' patient record cards, IPT cards, and IPT registers. Further, we assessed baseline characteristics as potential correlates of developing active TB during and after treatment and IPT completion using multivariable logistic regression.

### Results

From the data warehouse, 138,442 PLHIV were enrolled into ART during the study period and initiated 95,431 (68.9%) into IPT. We abstracted 4708 patients' files initiated on IPT, out of which 3891(82.6%) had IPT treatment outcomes documented, 4356(92.5%) had ever screened for TB at every clinic visit, and 4,243(90.1%) had documentation of TB screening on the IPT tool before IPT initiation. 3712(95.4%) of patients with documented IPT treatment outcomes completed their treatment. 42(0.89%) of the abstracted patients developed active TB,16(38.1%) during, and 26(61.9%) after completing IPT. Follow up for active TB at 6-month post-IPT completion was done for 2729(73.5%) of patients with IPT treatment

**Funding:** The author(s) received no specific funding for this work. Data has been generated as part of continual HIV/TB program monitoring and evaluation.

**Competing interests:** The authors have declared that no competing interests exist.

outcomes. Sex, Viral load suppression, and clinic type were associated with TB development ($p < 0.05$). Levels 4, 5, FBO, and private facilities and IPT prescription practices were associated with IPT completion ($p < 0.05$).

## Conclusion

IPT initiation stands at two-thirds of the PLHIV, with a high completion rate. TB screening practices were better during IPT than after completion. Development of active TB during and after IPT emphasizes the need for a keen follow up.

## Introduction

Tuberculosis (TB) is the most prevalent opportunistic infection among People Living with HIV (PLHIV) [1]. It remains the leading cause of death among the HIV population, accounting for one-in-three AIDS-related deaths [1]. Exposure to the *Mycobacterium tuberculosis* bacteria leads to TB infection, which can be controlled and remain inactive for years, but it can also progress to active TB [2]. Overall, the risk of developing active TB is 20 times more among PLHIV than those who do not have HIV infection [3]. The risk of active TB in HIV-infected persons continues to increase as HIV disease progresses, and immunity decreases [4]. The sustainable development goals (SDGs) and the World Health Organization (WHO) END TB Strategy aim to end the global TB epidemic by 2030 [5].

Kenya is listed by the World Health Organization (WHO) as among the 30 high burden TB countries currently facing the triple burden of TB, TB/HIV, and MDR TB with a TB prevalence of 558 per 100,000 population [6]. Despite the considerable investment done by the government and partners in TB care and prevention in the past 20 years, the disease is still the leading cause of death among PLHIV in the country [7]. According to the Kenya Tuberculosis Prevalence Survey 2016, half of all patients who fall ill to the disease go undiagnosed and untreated. The survey identified HIV as a significant risk factor, contributing to 17% of the overall TB burden [8]. The HIV co-infection rate among notified TB patients in Kenya is at 27% [8]. Kenya has a generalized HIV epidemic, which varies by geographical areas. According to the Kenya HIV 2018 estimates, the HIV prevalence among adults stands at 4.8% (Males 4.5%, Females 5.2%), translating to 1,493,400 (1,388,187 adults and 105,213 children) people living with HIV [9]. A third of PLHIV deaths are attributable to tuberculosis, a preventable and treatable disease [10].

In 1993, the WHO issued the first policy statement that recognized the efficacy of management of latent TB infection among PLHIV with isoniazid preventive therapy (IPT). WHO and Stop TB partnership in 2004 then developed and adopted an interim policy on the 3I's for HIV/TB collaborative activities. The 3Is covers the aspects of intensified TB case-finding and ensuring high-quality anti-tuberculosis treatment, initiation of TB prevention with IPT, and providing control of TB Infection in health-care facilities and congregate settings [11, 12]. In 2009, Kenya adopted the WHO 3I's and two more interventions that included: Immediate ART Therapy and Integration of TB and HIV—collectively dubbed the 5I's [13].

Isoniazid prophylaxis offers protection against TB for a range of 1 to more than 19 years, depending on the duration of isoniazid intake and the TB burden setting [14–18]. IPT for PLHIV was first recommended by WHO in 1998 and adopted by Kenya for rollout from 2015. After the implementation, the scale-up of IPT among PLHIV initially had a slow start but later peaked in 2016 as derived from the National Data Health Information system (S1 Fig). However, there is limited information on treatment outcomes, including limited published data

offering insights into potential gaps and strengths of the implementation of IPT in Kenya. To address this, we carried out this evaluation to determine the proportion of PLHIV initiated on IPT, assessed TB screening practices during and after IPT and IPT treatment outcomes. The findings will help inform the national HIV and TB programs and stakeholders on the progress of IPT implementation and help measure progress towards meeting TB control targets.

## Material and methods

### Study design

A retrospective study of PLHIV on ART and initiated on IPT was conducted using two data sources. We abstracted patient IPT uptake data for July 2018 and June 2018 period from the National data warehouse. In contrast, we obtained information on socio-demographic, TB screening practices, IPT initiation, follow up, and outcomes from selected health facilities' patient record cards, IPT cards, and IPT registers.

### Study setting

We conducted the study in 30 out of the 47 counties of Kenya. In each county, we selected representative facilities from both the urban and the rural setup with a different burden of TB/HIV. The 30 counties purposively and conveniently selected based on the ones with the highest TB caseload in Kenya. In Kenya, we perform TB screening for all new people entering the HIV program. The screening process involves a review of symptoms: looking for cough, fever, night sweats, weight loss, shortness of breath, sputum production and chest pain, a history of TB diagnosis or treatment, and a physical examination with a chest examination and lymph node exam. If there are any signs or symptoms of TB or an abnormal chest x-ray, a sputum exam is ordered—and the diagnostic process continues until TB has either been diagnosed or excluded. IPT is given for nine months to every client who screens negative for TB and who has not previously been on TB treatment.

The IPT screening program uses standard data collection forms, and the data are documented in the MoH forms and electronic medical records. The key endpoints recorded are either IPT completion or premature discontinuation (stopping before nine months).

### Study population

All PLHIV on ART and initiated on IPT between 1st July 2015 through 30th June 2018. The PLHIV get services at various levels of facility. The naming of levels depend on the service and the officer in charge of the facility; level 1, facilities run by certified medical clinical officers; level 2, facilities are run by clinical officers; level 3, small hospitals run by at least one doctor, clinical officers and nurses; level 4, hospitals that offer holistic services and are ran by a director who is a medic and at best a doctor by profession; level 5, county referral hospitals formerly the provincial hospitals. They are run by Chief Executive Officers who are medic by profession and have over 100 beds capacity for their in-patient; and level 6, teaching and Research referral hospitals.

### National data warehouse

The National Data Warehouse (NDW) is an integrated repository for HIV data from multiple Electronic Medical Records (EMR) and HIV Testing Services (HTS) applications in Kenya. The NDW stores rich de-identified individual-level data uploaded every month from all Health facilities in Kenya.

## Sample size determination

The required sample size was determined using a single population proportion formula n = (z2 * P (1-P)/ d2), and estimations for calculating sample size derived from findings of a study done in Nairobi, Kenya [19], study precision of 5% and confidence co-efficient of 1.96. We obtained a minimum sample size of (1,700 × 2) + 10% = 3,740.

## Sampling procedure

We randomly sampled 30 out of 47 counties. Further, the study purposely selected the county referral hospital, a sub-county referral hospital, a private hospital, and a faith-based facility from each of the sampled counties as an adequate representation of facility levels and types. The ultimate sampling unit from the facility was the Comprehensive Care Centers offering IPT services. From each sampling unit, we developed a sampling frame from the IPT patient registers and electronic medical registers where possible. The sample size of 3,740 was shared proportionally to size among the selected facilities that offer ART services in each county; hence approximately 32 patient files per facility were reviewed. Within the selected facilities, the allocated sample size was divided equally among the three years of the study period. We determined the random sampling interval of patient files by dividing the total number of patients started on IPT that year by the sample size allocated for the same year to get the sampling interval.

## Data collection and management

From the National data warehouse, we abstracted aggregate data for PLHIV initiated on IPT. Also, data on ART, socio-demographic and clinical characteristics for the same period were abstracted from patient files in facilities into Open Data Kit (ODK) using a standard patient-specific electronic tool based on the sampling criteria. The facility data abstraction tools were pre-tested in a level 4 facility. The data source was from the MOH recording tools in the health facilities, which included patient record card/file, ICF/IPT card (MOH257), and IPT registers. Specifically, at the facility, we extracted information on socio-demographic characteristics (age, sex), TB screening practices, IPT initiation, IPT follow up, and IPT outcomes (completed, stopped/discontinued, loss to follow-up, died, and transfer out). We also collected data on TB diagnosis during IPT and after IPT completion. The study team consisted of trained research assistants under the supervision of study coordinators. The data was backed-up regularly, and confidentiality maintained following national standards.

## Data analysis

Data was entered, cleaned, and analyzed using STATA version 15. We then calculated frequencies and proportions for categorical variables and appropriate measures of central tendencies and dispersion for continuous variables. Further, we compared patients' characteristics and clinical practices to IPT completion and other study outcomes using Pearson's chi-square test and relative risk as appropriate. Factors influencing the primary outcome, IPT completion, were assessed by logistic regression at both bi-variable and multivariable levels and effects presented as odds ratios (adjusted and unadjusted odds ratios) with their 95% confidence intervals. We considered factors with p<0.25 in the bivariate analysis for the multivariable levels. The level of significance was set at *P*-value <0.05.

## Ethical considerations

We maintained the confidentiality of the patient information collected by using codes for patients' identifiers and data stored in password-protected computers. The facility in charge

granted permission to assess the health facility tools for data abstraction. Ethical clearance was obtained from AMREF Ethical and Scientific Research Committee (AMREF- ESRC P531/ 2018).

## Results

### Study participants

A total of 138,442 and 95,431 (68.9%) PLHIV had been initiated on ART and IPT, respectively, as per data retrieved from the National reporting warehouse. A sample of 4708 patients files was abstracted, 3891(82.6%) had IPT treatment outcomes documented, while 817 (17.4%) did not have documented treatment outcomes. Of the 3891 initiated on IPT, 3712 (95.4%) completed their treatment, 97 (2.5%) stopped/discontinued treatment, 46 (1.2%) were lost to follow up, 22 (0.6%) transferred out and 14 (0.4%) died, 42 (0.89%) developed active TB with 26 (0.55%) being diagnosed after completing IPT (S2 Fig). The characteristics of the participants enrolled (N = 4708) are described in Table 1. Of these, 1423 (30.2%) were from level 2 facilities,

**Table 1. Characteristics of PLHIV who were initiated on IPT in Kenya, between 1st July 2015 and 30th June 2018.**

| Variables | Frequency | Percentage (%) |
|---|---|---|
| **Sex** | | |
| Female | 3159 | 67.1 |
| Male | 1549 | 32.9 |
| **Age Group** | | |
| 0–9 | 190 | 4.04 |
| 10–14 | 151 | 3.21 |
| 15–19 | 109 | 2.32 |
| 20–24 | 254 | 5.4 |
| 25+ | 4004 | 85.05 |
| **Patients on ART at Initiation of IPT** | | |
| Yes | 4525 | 96.1 |
| No | 183 | 3.9 |
| **Facility Level** | | |
| Level 2 | 1423 | 30.23 |
| level 3 | 1723 | 36.6 |
| level 4 | 1402 | 29.78 |
| level 5 | 160 | 3.4 |
| **Facility Ownership** | | |
| FBO | 1025 | 21.77 |
| GOK | 2845 | 60.43 |
| Private | 838 | 17.8 |
| **IPT Clinic** | | |
| CCC | 4123 | 87.57 |
| Inpatient | 28 | 0.59 |
| MCH/PMTCT | 257 | 5.46 |
| Outpatient clinic | 285 | 6.05 |
| TB clinic | 15 | 0.32 |

ART–antiretroviral therapy; FBO–Faith-based organization; GOK–Government of Kenya; IPT–isoniazid preventive therapy; CCC–comprehensive care clinic; MCH–maternal and child health; PMTCT–prevention of mother to child transmission; TB—tuberculosis

2845 (60.4%) were from government-run facilities. Among those enrolled on IPT, 4123 (87.6%) were in the CCC clinics. Females were 3159 (67.1%), 4004 (85.05%) were more than 25 years, and 4525 (96.1%) of the study participants were on ART at the initiation of IPT.

## IPT initiation and outcomes

Those who had ever been screened for TB at every clinic visit were 4356 (92.5%) (Table 2). Among PLHIV screened for TB, 2922 (67.1%) were females, while 3727 (85.6%) were in the age group 25 years and above. Compared to young adults 20–24 years old, the rest of the age groups were more likely to be routinely screened for TB (P<0.05), except for young adolescents 10–14 years old. Of the PLHIV routinely screened, 4,243 (90.1%) had the TB screening documented in the ICF tool before IPT initiation. Among those screened using the ICF tool, 293 (6.9%) were children <15 years, while 3950 (93.1%) were adults (≥15 years). The symptoms assessed in the screening for TB for as documented in the ICF tool for adults included cough (3944, 99.8%), fever (3917, 99.2%), weight loss (3940, 98.7%), and night sweats (3937, 99.7%). TB Symptoms assessed in children were cough (293, 100%), fever (291, 99.3%), weight loss (258, 88.1%), night sweats (251, 85.7%), failure to thrive (190, 64.8%) and lethargy (151, 51.5%).

**Table 2. Routine TB screening and use of ICF tool for screening of TB among adults and children living with HIV, Kenya, 2015–2018.**

| Variable | Frequency | Percent (%) |
|---|---|---|
| **Screened for TB routinely (n = 4356)** | | |
| **Sex** | | |
| Female | 2922 | 67.1 |
| Male | 1434 | 32.9 |
| **Age group (years)** | | |
| 0–9 | 174 | 4.0 |
| 10–14 | 137 | 3.1 |
| 15–19 | 102 | 2.3 |
| 20–24 | 216 | 5.0 |
| 25+ | 3727 | 85.6 |
| **ICF card used** | | |
| Yes | 4243 | 97.4 |
| No | 113 | 2.6 |
| **Symptom used for screening** | | |
| **Among adults (n = 3950)** | | |
| Cough | 3944 | 99.8 |
| Weight loss | 3940 | 99.7 |
| Night Sweat | 3937 | 99.7 |
| Fever | 3917 | 99.2 |
| **Among children (n = 293)** | | |
| Cough | 293 | 100 |
| Fever | 291 | 99.3 |
| Weight loss | 258 | 88.1 |
| Night Sweat | 251 | 85.7 |
| Failure to Thrive | 190 | 64.8 |
| Lethargy | 151 | 51.5 |

TB–tuberculosis; ICF–intensified case finding for TB

**Table 3. Characteristics for patients developing TB during and after IPT in Kenya, 2015–2018.**

| Category | Overall n (%) | TB Diagnosis | | P-value |
|---|---|---|---|---|
| | | During IPT n (%) | Post IPT n (%) | |
| **Age group (yrs)** | | | | |
| 0–9 | 4 (9.5) | 1(6.2) | 3 (11.5) | 0.5 |
| 10–14 | 1 (2.3) | 0 (0) | 1 (3.8) | |
| 15–19 | 3 (7.1) | 2 (12.5) | 1 (3.8) | |
| 20–25 | 2 (4.7) | 0 (0) | 2 (7.6) | |
| 25+ | 32 (76.1) | 13 (81.2) | 19 (73.0) | |
| **Sex** | | | | |
| Female | 24 (57.1) | 5 (31.2) | 19 (73) | 0.008 |
| Male | 18 (42.8) | 11(68.7) | 7 (26.9) | |
| **Viral load before IPT initiation** | | | | |
| Unsuppressed | 6 (14.2) | 1(6.2) | 5 (19.2) | 0.8 |
| Suppressed | 13 (30.9) | 3 (18.7) | 10 (38.4) | |
| Not Documented | 23 (54.7) | 12 (75) | 11 (42.3) | |
| **Viral load after IPT Initiation** | | | | |
| Unsuppressed | 7 (16.6) | 0 (0) | 7 (26.9) | 0.03 |
| Suppressed | 27 (64.2) | 12 (75) | 15 (57.6) | |
| Not Documented | 8 (19) | 4 (25) | 4 (15.3) | |
| **Facility type** | | | | |
| Faith-based | 13 (30.9) | 4 (25) | 9 (34.6) | 0.07 |
| Public | 26 (61.9) | 9 (56.2) | 17 (65.3) | |
| Private | 3 (7.1) | 3 (18.7) | 0 (0) | |
| **Routine screening** | | | | |
| Yes | 41 (97.6) | 15 (93.7) | 26 (100) | 0.2 |
| No | 1 (2.4) | 1 (6.3) | 0 | |
| **Clinic IPT initiated** | | | | |
| CCC | 39 (92.8) | 13 (81.2) | 26 (100) | 0.02 |
| OPD | 3 (7.2) | 3 (18.8) | 0 | |

TB–tuberculosis; IPT–isoniazid preventive therapy; CCC–comprehensive care clinics; OPD–out-patient department

Of the 42 (0.89%) PLHIV on IPT who developed active TB, 26 (0.55%) were diagnosed after completing IPT while 16 (0.34%) developed TB while on IPT. The median time of being diagnosed with TB after IPT initiation was 2.5 months (interquartile range: 1–5 months). Table 3 describes the characteristics of these patients developing TB. Fourteen participants who died while on IPT are described in Table 4 - nine were males, median age was 39 years (range 17–65), and none of them had TB diagnosis.

Of the 3712 patients who completed IPT, 2729 (73.5%) were followed up for TB status at six months post-IPT completion. The patient follow-up post-IPT became less frequent after that with 554 (14.9%) followed up at 12 months, 144 (3.9%) at 18 months, and 285 (7.7%) at 24 months as shown in Fig 1.

## Factors associated with IPT completion

Factors significantly associated with completion of IPT included health facility level 4 (aOR 0.50, CI 0.33–0.74; *P* = 0.001), level 5 (aOR 0.39, CI 0.17–0.90; *P* = 0.03); privately-owned facilities (aOR 0.61, CI 0.43–0.87; *P* = 0.006); faith-based organizations (aOR 1.65, CI 1.02–2.68,

Table 4. Characteristics for patients who died during IPT in Kenya, 2015–2018 (n = 14).

| Variables | Frequency | Percentage |
|---|---|---|
| **Sex** | | |
| Female | 5 | 35.7 |
| Male | 9 | 64.3 |
| **County** | | |
| Baringo | 3 | 21.4 |
| Homa_Bay | 1 | 7.14 |
| Isiolo | 5 | 35.7 |
| Kitui | 1 | 7.1 |
| Makueni | 1 | 7.1 |
| Narok | 1 | 7.1 |
| Trans_Nzoia | 1 | 7.1 |
| Uasin_Gishu | 1 | 7.1 |
| **Viral Load Results after IPT** | | |
| Detectable > 1000 Copies/ml | 2 | 14.3 |
| LDL < 1000 Copies/ml | 1 | 7.1 |
| Not Documented | 12 | 85.7 |
| **On Pyridoxine** | | |
| No | 1 | 7.1 |
| Yes | 8 | 57.1 |
| Not Documented | 5 | 35.7 |
| **IPT Prescription** | | |
| Monthly | 13 | 92.9 |
| Every 4 months | 1 | 7.1 |

IPT–isoniazid preventive therapy; LDL–lower detectable limit

P = 0.04) and IPT prescription practices (monthly, 2 months, 3 months and above 4 months) (Table 5).

## Discussion

About two-thirds of the patients were initiated on IPT for the period under review, indicating that IPT uptake can be scaled up nationally. The reason for this in this setting was the adoption of IPT policy in the National HIV guidelines 2011 [20]; Communication guidance addressed to health care workers on IPT initiation from the Ministry; the collaboration of the National HIV and TB programs; adoption of the Rapid results Initiatives which entailed IPT target setting for the counties and continuous engagement with the civil society organizations. Other studies have demonstrated high IPT uptake alongside other factors [21–24]. We did not probe into why the rest were not on IPT, though based on unpublished programmatic data, this is attributable to ineligibility status due to contraindications for IPT: active TB at the time of ART initiation or predisposing Liver disease. However, a study in an urban health centre in the capital Nairobi reported a 77% IPT uptake [19] amongst adults PLHIV who had been in care for at least six months and regardless of ART status. Though our study was focusing on only those on ART, we do not think this differs much as the country's policy has been the initiation of IPT regardless of ART status.

Kenya adopted routine TB screening for PLHIV at every clinic visit by use of the ICF tool in 2011 [20, 25] to accommodate for pediatrics and Adults. This has recently been reinforced

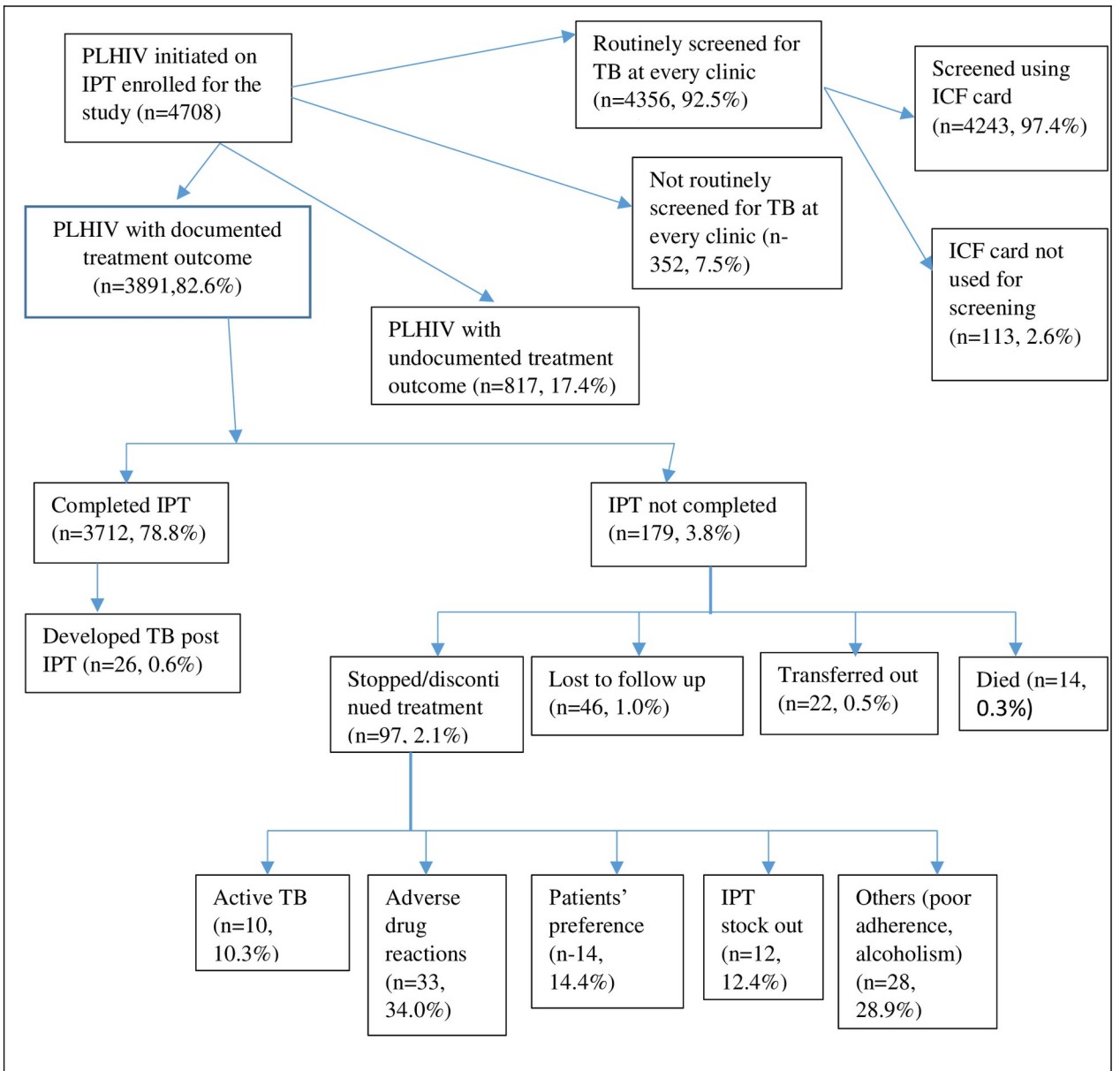

**Fig 1. Flow diagram of persons living with HIV who were initiated on isoniazid preventive therapy in Kenya between 1st July 2015 and 30th June 2018.** IPT–Isoniazid Preventive Therapy; PLHIV–People Living with HIV; ART–Antiretroviral Therapy; TB–Tuberculosis; ICF–intensified case finding.

in the recent National ARV Guideline, 2018 edition that states, screening PLHIV actively for TB by use of the ICF tool at every clinical encounter allows for the provision of TB preventive therapy services among those eligible [26]. We found out that a majority of the PLHIV were regularly screened for the four TB symptoms. Our study is comparable to a previous study in western Kenya that found much higher screening coverage. Partly, we attribute this to the increased focus on the quality of care interventions among PLHIV in the past two years [27]. These findings revealed a better uptake of screening for TB, unlike in the UK, where a clinical audit found that less than a quarter of the PLHIV routinely screened for TB [28]. Also, routine screening for TB was applied equally among males and females, but this differed among the

**Table 5. Factors associated with IPT completion among PLHIV in Kenya, 2015–2018.**

| Variables | uOR | (95% CI) | p-value | aOR | 95% CI | p-value |
|---|---|---|---|---|---|---|
| **Where IPT was prescribed (n = 3,891)** | | | | | | |
| CCC | Ref | | | | | |
| In-Patient | 1.03 | (0.14–7.68) | 0.9 | 0.79 | (0.1–5.96) | 0.8 |
| MCH/PMTCT | 0.55 | (0.33–0.92) | **0.02** | 0.66 | (0.39–1.12) | 0.1 |
| Out-Patient | 1.4 | (0.65–3.02) | 0.4 | 1 | (0.46–2.21) | 0.9 |
| TB Clinic | 0.47 | (0.06–3.68) | 0.5 | 0.24 | (0.03–1.98) | 0.2 |
| **Facility level (n = 3,891)** | | | | | | |
| level 2 | Ref | | | | | |
| level 3 | 0.95 | (0.62–1.46) | 0.8 | 0.98 | (0.63–1.51) | 0.9 |
| level 4 | 0.48 | (0.33–0.71) | **0** | 0.5 | (0.33–0.74) | **0.001** |
| level 5 | 0.56 | (0.26–1.23) | **0.1** | 0.39 | (0.17–0.90) | **0.03** |
| **Type of Facility(n = 3,891)** | | | | | | |
| GoK | Ref | | | | | |
| FBO | 1.71 | (1.06–2.75) | **0.03** | 1.65 | (1.02–2.68) | **0.04** |
| Private | 0.59 | (0.42–0.83) | **0.002** | 0.61 | (0.43–0.87) | **0.006** |
| **IPT duration Prescription (n = 3,891)** | | | | | | |
| < month | Ref | | | | | |
| Monthly | 1.83 | (0.96–3.5) | **0.07** | 2.13 | (1.10–4.13) | **0.03** |
| Every 2 m | 3.58 | (1.71–7.48) | **0.001** | 4.22 | (1.99–8.95) | **<0.01** |
| Every 4 m | 2.48 | (1.18–5.19) | **0.02** | 3.01 | (1.39–6.52) | **0.005** |
| Above 4 m | 7.12 | (1.55–32.83) | **0.01** | 7.65 | (1.64–35.7) | **0.01** |
| **Routine screen(n = 3,891)** | | | | | | |
| Yes | Ref | | | | | |
| No | 0.65 | (0.38–1.10) | 0.1 | | | |
| **Sex(n = 3,891)** | | | | | | |
| Female | Ref | | | | | |
| Male | 0.81 | (0.59–1.10) | 0.2 | | | |
| **Age-group (n = 3,891)** | | | | | | |
| 0–9 | Ref | | | | | |
| 10–14. | 1.33 | (0.47–3.76) | 0.6 | | | |
| 15–19 | 0.91 | (0.32–2.59) | 0.9 | | | |
| 20–25 | 1.53 | (0.61–3.87) | 0.4 | | | |
| 25+ | 1.45 | (0.75–2.82) | 0.3 | | | |
| **Viral load before IPT (n = 1,596)** | | | | | | |
| Suppressed | Ref | | | | | |
| Not Suppressed | 1.14 | (0.48–2.71) | 0.8 | | | |
| **Viral load After IPT (n = 3348)** | | | | | | |
| Suppressed | Ref | | | | | |
| Not Suppressed | 0.61 | (0.35–1.08) | 0.09 | | | |
| **ART initiation (n = 3891)** | | | | | | |
| Before | Ref | | | | | |
| After | 0.59 | (0.31–1.15) | 0.1 | | | |

uOR–unadjusted Odds Ratio; aOR–adjusted Odds Ratio; CI—Confidence interval; CCC–comprehensive care clinics; MCH/PMTCT–maternal and child health/ prevention of mother to child transmission; TB–tuberculosis; GoK–the government of Kenya; FBO–faith-based organization; IPT–isoniazid preventive therapy; ART–antiretroviral therapy

**In bold**–statistically significant at P<0.05

age groups with the young adults aged 20–24 years having lower chances of being routinely screened when compared to the rest. The reason for these findings is not apparent but are similar to those reported in a study in western Kenya [27]. In children, the additional symptoms required for screening, such as failure to thrive, and lethargy were sub-optimally documented, showing missed opportunities for actively looking for TB among them, which has been evidenced to detect TB cases [29].

The IPT completion rates in Kenya were above 95%, which is higher compared to other African countries, where it ranges from 70–94% [30–32]. The excellent results are attributable to the adoption and implementation of policy guidelines on IPT in Kenya, integration of tuberculosis and HIV services, sensitization of the health care workers on the importance of IPT among PLHIV and mechanisms to support treatment adherence in HIV care clinics. In Kenya, adverse drug reactions were the leading cause of IPT non-completion. South Africa and Zimbabwe reported similar findings where 3.8% and 7%, respectively discontinued IPT due to side effects [31, 33].

Predictors of IPT completion in this study were found to the level of facility with level 4, 5, and private health facilities having lower completion rates. This could be due to the no or partial interaction of TB/HIV services, thus challenges of quality of TB/HIV care. A study in Swaziland found high IPT completion rates despite the level of facility delivering it [32]. The need for enhanced adherence counseling and active tracing amongst clients on IPT has proved to be vital for successful completion of IPT [31].

The study findings of those who developed active TB as an outcome was documented with two-thirds of the patients' post IPT getting infected than during IPT. A representative retrospective study in Myanmar involving 3377 participants, also seems to have a similar comparable picture of two-thirds of the patients developing active TB post-IPT than during IPT [34]. This emphasizes the need for continuous TB screening during and after IPT and adherence counseling while on medication.

Our findings showed that TB screening guidelines were followed up well during the administration of IPT, but after completion of IPT, there was laxity. Only about two-thirds of the patients who completed IPT were followed up for active TB status at six months post-IPT completion. The patient follow-up post-IPT became less frequent after that with less than a quarter followed up at subsequent months. According to national guidelines, follow up of PLHIV on IPT should be conducted monthly during IPT with rescreening for TB at every visit to help address the adverse drug events and to detect any signs of active TB for initiation of early treatment. The guidelines ensure patients who have active TB disease do not end up developing drug resistance TB later in life. Also, the patients should be followed up for the next two years post IPT to ensure active TB is detected early.

## Study limitation

Programmatic data has inadequate data on TB screening for patients who only come for prescription refills, and symptom screening conducted as part of routine clinical services are not available for verification. Additionally, under these normal practice conditions, we were unable to obtain complete data on all patients. Due to the low numbers of PLHIV on IPT who developed TB, it was not possible to establish the relationship between patient characteristics and TB as an outcome. Further, over 60% of patients who developed TB did not have a viral load, hence this could not be evaluated during the characterization.

## Conclusion

Kenya MoH initiated two-thirds of PLHIV on IPT during the study period, and the completion rate was very high. TB screening practices were not up to the standards for all the PLHIV,

and a few patients were still diagnosed with active TB during IPT uptake and also post-IPT completion. There remains a gap in TB screening for the PLHIV, and among those screened, the ICF tool is not uniformly applied. IPT completion rates among HIV infected patients were demonstrated to be high. Routine TB screening while on IPT was better than after IPT completion.

There is a need to strengthen TB/HIV integration to ensure all PLHIV are routinely screened for TB and sensitize the health care workers on the use of the ICF tool for screening. More emphasis is required on documentation for IPT clients to reduce the proportion of clients with no treatment outcomes recorded. Quality of care across all health facilities should be enhanced to ensure similar treatment outcomes not based on the level of ownership. The study recommends further evaluation of factors associated with the development of TB during and after IPT completion. Also, we need to assess the quality of screening of TB among PLHIV and conduct routine data audits on ICF.

## Supporting information

**S1 Fig. The use of IPT among PLHIV from 2011 to 2018 in Kenya.** The peak in 2015–16 is attributed to both the development of IPT policy by the Ministry of health and 100-days rapid results initiative (RRI).
(DOCX)

**S2 Fig. Frequency of TB screening post IPT completion among PLHIV, Kenya, 2015–2018.**
(DOCX)

**S1 Dataset.**
(XLSX)

## Acknowledgments

This 2019 report findings on the assessment of the outcomes of isoniazid preventive therapy among people living with HIV in Kenya is through collaborative efforts of individuals and institutions led by Ministry of Health through the Division of National AIDS and STI Control Program (NASCOP) and Division of National Tuberculosis, Leprosy and Lung Disease Program (NTLD-P). We thank the study coordinator from MOH Moseti Makori and the IPT study team members that consisted of a team from Division of NASCOP (Steve Ambune and Evans Imbuki), Division of NLTD-P (Kiogora Gatimbu, Newton Omale, Richard Kiplimo, and Martin Githiomi), MOH (Kigen Bartilol, Maureen Kamene, George Githuka, Stephen Muleshe), NPHL (Josephine Wahogo), KEMRI (Jane Ong' ang' o), KNH (Margaret Oluka), CHS (Lorraine Mugambi-Nyaboga, Evelyne Ng'ang'a and Wandia Ikua), WHO (Hillary Kipruto) and the Global CHAI Team.

## Author Contributions

**Conceptualization:** Muthoni Karanja, Leonard Kingwara, Philip Owiti, Enos Masini, Elizabeth Onyango.

**Data curation:** Muthoni Karanja, Elvis Kirui, Faith Ngari, Richard Kiplimo, Maurice Maina, Catherine Ngugi.

**Formal analysis:** Muthoni Karanja, Leonard Kingwara, Philip Owiti, Richard Kiplimo, Maurice Maina.

**Funding acquisition:** Muthoni Karanja.

**Investigation:** Muthoni Karanja, Philip Owiti, Maurice Maina, Elizabeth Onyango.

**Methodology:** Muthoni Karanja, Leonard Kingwara, Philip Owiti, Faith Ngari, Richard Kiplimo, Enos Masini, Elizabeth Onyango, Catherine Ngugi.

**Project administration:** Muthoni Karanja, Maurice Maina.

**Resources:** Muthoni Karanja, Catherine Ngugi.

**Supervision:** Maurice Maina.

**Writing – original draft:** Muthoni Karanja, Leonard Kingwara, Elvis Kirui, Faith Ngari, Richard Kiplimo.

**Writing – review & editing:** Muthoni Karanja, Leonard Kingwara, Philip Owiti, Elvis Kirui, Faith Ngari, Richard Kiplimo, Maurice Maina, Enos Masini, Elizabeth Onyango, Catherine Ngugi.

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
