## [Decision Letter · Decision Letter 0]

15 Jun 2020

PONE-D-20-15776

Outcomes of Isoniazid Preventive Therapy among people living with HIV in Kenya: A retrospective study of routine health care data

PLOS ONE

Dear Mr Kingwara,

Thank you for submitting your manuscript to PLOS ONE. After careful consideration, we feel that it has merit but does not fully meet PLOS ONE’s publication criteria as it currently stands. Therefore, we invite you to submit a revised version of the manuscript that addresses the points raised during the review process.

We look forward to receiving your revised manuscript.

Kind regards,

Professor Kwasi Torpey, MD PhD MPH

Academic Editor

PLOS ONE

Journal Requirements:

This 2019 report findings on the assessment 293 of the outcomes of isoniazid preventive therapy among

people living with HIV in Kenya is through collaborative efforts of individuals and institutions led by

Ministry of Health through the Division of National AIDS and STI Control Program (NASCOP) and

Division of National Tuberculosis, Leprosy and Lung Disease Program (NTLD-P). We thank the study

co-ordinator from MOH Moseti Makori and the IPT study team members that consisted of a team

from Division of NASCOP (Steve Ambune and Evans Imbuki), Division of NLTD-P (Kiogora Gatimbu,

Newton Omale, Richard Kiplimo, and Martin Githiomi), MOH (Kigen Bartilol, Maureen Kamene,

George Githuka, Stephen Muleshe), NPHL (Josephine Wahogo), KEMRI (Jane Ong’ang’o), KNH

(Margaret Oluka ), CHS (Lorraine Mugambi-Nyaboga, Evelyne Ng`ang`a and Wandia Ikua),

WHO (Hillary Kipruto) and the Global CHAI Team. Special appreciation goes to Global Fund for

AIDS, TB and Malaria for the financial support in the implementation of the assessment.

The author(s) received no specific funding for this work. Data has been generated as

part of continual HIV/TB program monitoring and evaluation

Additional Editor Comments (if provided):

Please review all references and ensure it is complete and consistent with the journal requirements particularly references #2,7,8,9,14,15,16,19, 21, 22,24, 27, 28, 29, 30

Reviewers' comments:

Reviewer's Responses to Questions

**Comments to the Author**

1. Is the manuscript technically sound, and do the data support the conclusions?

Reviewer #1: Yes

Reviewer #2: Yes

2. Has the statistical analysis been performed appropriately and rigorously? 

Reviewer #1: Yes

Reviewer #2: Yes

3. Have the authors made all data underlying the findings in their manuscript fully available?

Reviewer #1: No

Reviewer #2: Yes

4. Is the manuscript presented in an intelligible fashion and written in standard English?

Reviewer #1: Yes

Reviewer #2: Yes

5. Review Comments to the Author

Reviewer #1: General comments

The paper addressed one of the strategies, of clinical and public health significance aiming to end the local and global TB epidemic. Isoniazid preventive therapy (IPT) is effective in reducing the incidence of active TB among HIV-infected people and other people at risk.

The introduction is well written and the paragraphs follow a logical flow of concepts. The methodology section is detailed, clear and easy to follow. The text results are well arranged but exhaustive of the tables and figures; numerical values are crowded in the text. In the discussion, the authors have discussed the results of other studies more than that of the current study findings.

In conclusion, the paper is relevant for clinical practice implications. However, there some areas need necessary corrections to improve the manuscript.

Specific comments:

1. Abstract:

On page I, line 16; the word "tuberculosis" is non-specific to the context. It should be written "active tuberculosis" or "tuberculosis disease". The authors should change this throughout the manuscript. In principle, IPT aims to reduce the risk of developing active TB or reactivation of latent TB infection in the population at risk such as people living with HIV.

2. Introduction:

On page 4, line 72; "Isoniazid prophylaxis taken for six months offers protection against TB for at least two years". No fixed time duration for the protection against developing active TB after completion of IPT. There is a great variation between studies in terms of design, setting, duration of the IPT, characteristics of the participants/population etc. Some studies have reported no protection, others 7yrs to 20 yrs etc. Therefore, the authors should state a range of time duration and cite the appropriate references. Below are some useful references.

o Comstock GW, Baum C, Snider DE., Jr Isoniazid prophylaxis among Alaskan Eskimos: a final report of the Bethel isoniazid studies. Am Rev Respir Dis. 1979; 119:827–30.

o Golub JE, Cohn S, Saraceni V, et al. Long-term protection from isoniazid preventive therapy for tuberculosis in HIV-infected patients in a medium-burden tuberculosis setting: the TB/HIV in Rio (THRio) study. Clin Infect Dis. 2015;60(4):639‐645. doi:10.1093/cid/ciu849

3. Methods:

On page 7, line 130; "with p<0.25 in the bivariate analysis were considered for multiple regression". what/which reference did you use to decide including a variable with p<0.25 in the multivariate analysis?

4. Results:

The results (numerical values) in the text are overcrowded. The authors should present striking/important findings in the text.

On page 8 lines 146, 147, and 152; and page 9 lines 165, 167, and 176-177: the referenced tables (table 1, table 2, table 3, table 4, and table5) and figures (figure 1, figure 2, and figure 3) are highlighted in blue colour; and printouts appear faint. The authors should use front colour option – automatic/black.

Table 2: no relevance to include a variable "symptoms for screening" in this table. ICF for TB can be described in the methods section - data collection.

Table 4: what is the significance of this table? I think you can summarize it in one to two sentences or attach as a supplementary table.

Table 5: decongest columns 4 by deleting a blank column 2. Why the variables: routine screening, sex, viral load after IPT, and ART initiation with p<0.25 were not included in the multivariate analyses? If there were multicollinearity, please produce the evidence. In case of sparse or missing data, please elaborate in the methods section or state at the footnote.

5. Discussion:

The authors should discuss the findings of the current study more than the findings of other similar studies. For example, paragraph 3 on page 17 lines 234-240 is well discussed; arguments are based on local (Kenya) evidence and compared with relevant findings from other studies. Revise the other paragraphs.

6. References:

Revise all of the references and write according to the Journal guideline. For example, the format for the following references is incorrect: references number 2,3,5,6,8-12 etc.

Reviewer #2: Summary

The authors conducted a retrospective review of records of people living with HIV (PLHIV) to determine the outcomes of isoniazid preventive (IPT) TB. The manuscript addresses a very important gap in the literature. Date are limited on the performance of the IPT cascade from high TB burden settings like Kenya. the authors report that two thirds of PLHIV on ART were initiated on IPT 95% of whom completed the six-month course.

Abstract

The abstract is well written and is a good summary of the manuscript. However, the first statement of the results which sounds as data collected outside of the described study. Because this statement is included in the main manuscript and contributes substantially to the conclusions I will elaborate on this comment in the results section of the manuscript.

Introduction

Paragraph 2 – the first statement refers to the TB burden in Kenya using TB incidence from WHO estimates. Since Kenya recently completed a TB prevalence survey and the results are available in published literature they should consider using this instead or in addition to the WHO estimates. Also when discussing the burden of TB/HIV co-infection in Kenya they refer to ministry of health reports. They should consider citing other analyses of Kenya TB surveillance data which have been published in peer reviewed journals as well.

Paragraph 3 – the authors have provided a list of specific objectives which could perhaps be summarized into fewer sentences.

Methods

Study setting: The authors should consider providing a more detailed summary of their study setting to enable readers to understand the study context. This could include information on how TB/HIV services are organized in Kenya especially IPT provision. What criteria was used to select the 30 counties that were included in this study?

Study population: "All PLHIV in care and initiated on IPT between 1st July 2015 through 30th June 2018"

This statement needs some clarification the authors included only clients on ART from specific facilities in specific counties. Also, before the implementation of test and treat approach to ART initiation, patients on care were not always on ART. Perhaps the word "care" should be replaced with "ART".

Data collection and management -

"The study team abstracted aggregate data for patients initiated

on IPT and PLHIV on ART" – this statement makes it sound as though the patients on IPT was a separate inclusion criteria from PLHIV on ART. Did the authors mean to say " PLHIV on IPT and ART"? Please revise this statement to reflect the study population.

Results

1) The currently described methods do not support this result: "We enrolled 138,442 PLHIV into ART during the study period and initiated 95,431 (68.9%) into IPT as per data retrieved from the National reporting warehouse." In fact, within this result the authors declare a different data source from what is described in the methods. If the authors wish to keep this result they need to revise the methods and include how they collected this data. Otherwise it is confusing how out of a sample size of roughly 3700 PLHIV a result of over 138,000 PLHIV is obtained. Also clarify whether these 138, 442 were from the entire country or from the study sites alone.

2) The presentation of results could be improved by having it structured into sections – description of study participants, IPT initiation and outcomes and factors associated with IPT completion

3) The authors should consider improving the logical flow of the results – its confusing to start with IPT outcomes, pedal back to screening which is a prerequisite to IPT initiation and back to outcomes (death and active TB)

4) Facility levels (2, 4, 5) are mentioned but have not been defined in the methods. Perhaps this also applies to facility ownership.

5) Figures and tables – the authors have included 3 figures and five tables. Perhaps some of these could be included as supplementary materials

Discussion and conclusions

Reference to the proportion of PLHIV on ART initiating IPT should be reviewed based on my earlier comment about this result being ectopic.

The authors should include their perspective on why IPT initiation was lower in higher level and private facilities.

References

1. The authors have relied a lot on non-peer reviewed sources (Ministry of Health and WHO). They should strive to rely more on peer reviewed sources if they are available.

2. The formatting of the references should be revised. Author names of institutions are abbreviated yet what these acronyms stand for is not spelt out (example – MOH).

6. PLOS authors have the option to publish the peer review history of their article (what does this mean?). If published, this will include your full peer review and any attached files.

Reviewer #1: Yes: Festo K. Shayo

Reviewer #2: Yes: Dickens Otieno Onyango

---

## [Author Response · Author response to Decision Letter 0]

28 Oct 2020

Dear Editor, 

Sincerest thanks for your response and reviewers' comments on our manuscript. We sincerely apologize for the great time it has taken us to respond to these comments and hope that a revised version of the manuscript will still be considered by Journal of Plos One. We have modified the paper in response to the extensive and insightful reviewer comments. In the abstract section, we clarified the methods section to give the manuscript proper angulation and clarity to address the reviewer's comments adequately. Also, we have rewritten sections of the manuscript, and we hope that this complies with the referee's remarks. We will thus respond to the comments point counter, as indicated below. 

1. Abstract

The abstract is well written and is a good summary of the manuscript. However, the first statement of the results which sounds as data collected outside of the described study. Because this statement is included in the main manuscript and contributes substantially to the conclusions, I will elaborate on this comment in the results section of the manuscript. 

We have clarified by reviewing the abstract as well as the methodology section to indicate that there are two sources of data used in the study (National data warehouse and patient files)

2. Introduction 

Paragraph 2 – the first statement refers to the TB burden in Kenya using TB incidence from WHO estimates. Since Kenya recently completed a TB prevalence survey and the results are available in published literature they should consider using this instead or in addition to the WHO estimates. Also when discussing the burden of TB/HIV co-infection in Kenya they refer to ministry of health reports. They should consider citing other analyses of Kenya TB surveillance data which have been published in peer-reviewed journals as well. 

We have revised the references to indicate publications from TB prevalence survey "Enos, M. et al. Kenya tuberculosis prevalence survey 2016: Challenges and opportunities of ending TB in Kenya. PLoS One 13, e0209098 (2018)”. Additionally, references to non-published program data have been removed and replaced with publications referencing TB prevalence survey and Kenya HIV/AIDS estimates "Mbithi, A. et al. Tuberculosis and HIV at the National Level in Kenya: Results From the Second Kenya AIDS Indicator Survey. doi:10.1097/QAI, Ministry of Health Republic of Kenya, Division of Tuberculosis, L. and L. D. Kenya TB Survey Report 2016. (2016) and the Kenya National AIDS Control Council (NACC). KENYA HIV ESTIMATES 2018. (2018)"

3. Paragraph 3 – the authors have provided a list of specific objectives which could perhaps be summarized into fewer sentences. 

We have revised the listing of all the four objectives into one sentence that read. “To determine the proportion of PLHIV in care initiated on IPT; to assess TB screening practices before and after IPT and to find out IPT treatment outcomes”

4. Methods

Study setting: The authors should consider providing a more detailed summary of their study setting to enable readers to understand the study context. This could include information on how TB/HIV services are organized in Kenya especially IPT provision. What criteria was used to select the 30 counties that were included in this study? We have revised the study setting chapter to give a clear picture of the IPT practices in Kenya. Further, in the same chapter, we have clearly documented how the 30 out of the 47 counties were arrived at.

5. Study population: "All PLHIV in care and initiated on IPT between 1st July 2015 through 30th June 2018" This statement needs some clarification the authors included only clients on ART from specific facilities in specific counties. Also, before the implementation of test and treat approach to ART initiation, patients on care were not always on ART. Perhaps the word "care" should be replaced with "ART".We acknowledge the comments and have revised it to read All PLHIV on ART and initiated on IPT between 1st July 2015 through 30th June 2018. 

6. Data collection and management - 

"The study team abstracted aggregate data for patients initiated

on IPT and PLHIV on ART" – this statement makes it sound as though the patients on IPT was a separate inclusion criterion from PLHIV on ART. Did the authors mean to say " PLHIV on IPT and ART"? Please revise this statement to reflect the study population.

We acknowledge, and the sentence has been revised to read All PLHIV on ART and initiated on IPT between 1st July 2015 through 30th June 2018. 

7. Results

1) The currently described methods do not support this result: "We enrolled 138,442 PLHIVfig into ART during the study period and initiated 95,431 (68.9%) into IPT as per data retrieved from the National reporting warehouse." In fact, within this result the authors declare a different data source from what is described in the methods. If the authors wish to keep this result they need to revise the methods and include how they collected this data. Otherwise it is confusing how out of a sample size of roughly 3700 PLHIV a result of over 138,000 PLHIV is obtained. Also clarify whether these 138, 442 were from the entire country or from the study sites alone. 

We have clarified this by adding a sub-chapter on the method section on Data warehouse that clearly describe how these data were obtained. Additionally, we have clarified this by adding the sentence "From the National data warehouse, we abstracted aggregate data for PLHIV initiated on IPT. Also, data on ART, socio-demographic and clinical characteristics for the same period were abstracted from patient files in facilities into Open Data Kit(ODK) using a standard patient-specific electronic tool based on the sampling criteria"

8. The presentation of results could be improved by having it structured into sections – description of study participants, IPT initiation and outcomes and factors associated with IPT completion

We acknowledge the comments. For angulation, we have split the results section Into three: (1)Study participants (2)IPT initiation and outcomes and (3)factors associated with IPT completion

9. The authors should consider improving the logical flow of the results – its confusing to start with IPT outcomes, pedal back to screening which is a prerequisite to IPT initiation and back to outcomes (death and active TB). The entire results section have been revised Into three sub-sections to Improve angulation

10. Facility levels (2, 4, 5) are mentioned but have not been defined in the methods. Perhaps this also applies to facility ownership.

We have acknowledge the comments and have defined the meaning of all levels In the methodology section

11. Figures and tables – the authors have included 3 figures and five tables. Perhaps some of these could be included as supplementary materials

Fig 1 and Fig 2 has been moved to the supplemental material.

12. Discussion and conclusions

Reference to the proportion of PLHIV on ART initiating IPT should be reviewed based on my earlier comment about this result being ectopic. We acknowledge the comment. This has been revised In line with comment number 5 and 6

13. The authors should include their perspective on why IPT initiation was lower in higher level and private facilities. Our results do not indicate a lower IPT initiation in higher-level and private facilities. However, I think the reviewer meant IPT completion. If so, the author's perspective has been given in the discussion section as "Predictors of IPT completion in this study were found to the level of facility with level 4, 5, and private health facilities having lower completion rates. This could be due to the no or partial interaction of TB/HIV services, thus challenges of quality of TB/HIV care."

14. References

The authors have relied a lot on non-peer-reviewed sources (Ministry of Health and WHO). They should strive to rely more on peer-reviewed sources if they are available. All the non-peer-reviewed sources have been replaced

15. The formatting of the references should be revised. Author names of institutions are abbreviated yet what these acronyms stand for is not spelt out (example – MOH). We have acknowledged and revised

16. Reviewer #1: General comments

The paper addressed one of the strategies of clinical and public health significance, aiming to end the local and global TB epidemic. Isoniazid preventive therapy (IPT) is effective in reducing the incidence of active TB among HIV-infected people and other people at risk.

The introduction is well written and the paragraphs follow a logical flow of concepts. The methodology section is detailed, clear and easy to follow. The text results are well arranged but exhaustive of the tables and figures; numerical values are crowded in the text. In the discussion, the authors have discussed the results of other studies more than that of the current study findings.

In conclusion, the paper is relevant for clinical practice implications. However, there some areas need necessary corrections to improve the manuscript.

Specific comments:

Abstract:

On page I, line 16; the word "tuberculosis" is non-specific to the context. It should be written "active tuberculosis" or "tuberculosis disease". The authors should change this throughout the manuscript. In principle, IPT aims to reduce the risk of developing active TB or reactivation of latent TB infection in the population at risk such as people living with HIV. We have noted and revised the same in the entire manuscript

17. Introduction:

On page 4, line 72; "Isoniazid prophylaxis taken for six months offers protection against TB for at least two years". No fixed time duration for the protection against developing active TB after completion of IPT. There is a great variation between studies in terms of design, setting, duration of the IPT, characteristics of the participants/population etc. Some studies have reported no protection, others 7yrs to 20 yrs etc. Therefore, the authors should state a range of time duration and cite the appropriate references. Below are some useful references.

o Comstock GW, Baum C, Snider DE., Jr Isoniazid prophylaxis among Alaskan Eskimos: a final report of the Bethel isoniazid studies. Am Rev Respir Dis. 1979; 119:827–30.

o Golub JE, Cohn S, Saraceni V, et al. Long-term protection from isoniazid preventive therapy for tuberculosis in HIV-infected patients in a medium-burden tuberculosis setting: the TB/HIV in Rio (THRio) study. Clin Infect Dis. 2015;60(4):639‐645. doi:10.1093/cid/ciu849. We acknowledge the concern and this has been clarified as "Isoniazid prophylaxis offers protection against TB for a range of 1 to more than 19 years, depending on the duration of isoniazid intake and the TB burden setting." References on the same have been added.

18. Methods:

On page 7, line 130; "with p<0.25 in the bivariate analysis were considered for multiple regression". what/which reference did you use to decide including a variable with p<0.25 in the multivariate analysis?. We acknowledge we are using the Cohen (1988) which suggests an R2 values for endogenous latent variables to be assesed as follows: 0.26 (substantial), 0.13 (moderate), 0.02 (weak)

19. Results:

The results (numerical values) in the text are overcrowded. The authors should present striking/important findings in the text.

On page 8 lines 146, 147, and 152; and page 9 lines 165, 167, and 176-177: the referenced tables (table 1, table 2, table 3, table 4, and table5) and figures (figure 1, figure 2, and figure 3) are highlighted in blue colour; and printouts appear faint. The authors should use front colour option – automatic/black.

Table 2: no relevance to include a variable "symptoms for screening" in this table. ICF for TB can be described in the methods section - data collection.

Table 4: what is the significance of this table? I think you can summarize it in one to two sentences or attach as a supplementary table.

Table 5: decongest columns 4 by deleting a blank column 2. Why the variables: routine screening, sex, viral load after IPT, and ART initiation with p<0.25 were not included in the multivariate analyses? If there were multicollinearity, please produce the evidence. In case of sparse or missing data, please elaborate in the methods section or state at the footnote.

Table 5 has been replaced, and the entire results section has been reviewed to make it readable. 

Discussion:

The authors should discuss the findings of the current study more than the findings of other similar studies. For example, paragraph 3 on page 17 lines 234-240, is well discussed; arguments are based on local (Kenya) evidence and compared with relevant findings from other studies. Revise the other paragraphs. The discussion section has been reviewed as evidenced in the track and clean copy.

6. References:

Revise all of the references and write according to the Journal guideline. For example, the format for the following references is incorrect: references number 2,3,5,6,8-12 etc. References have been revised to the PLoS one manuscript reference required format.

Thank you for your consideration of this manuscript.

---

## [Editor Report · Decision Letter 1]

2 Nov 2020

Outcomes of Isoniazid Preventive Therapy among people living with HIV in Kenya: A retrospective study of routine health care data

PONE-D-20-15776R1

Dear Mr Kingwara,

We’re pleased to inform you that your manuscript has been judged scientifically suitable for publication and will be formally accepted for publication once it meets all outstanding technical requirements.

Kind regards,

Kwasi Torpey, MD PhD MPH

Academic Editor

PLOS ONE
---

## [Editor Report · Acceptance letter]

14 Nov 2020

PONE-D-20-15776R1 

Outcomes of Isoniazid Preventive Therapy among people living with HIV in Kenya: A retrospective study of routine health care data 

Dear Dr. Kingwara:

I'm pleased to inform you that your manuscript has been deemed suitable for publication in PLOS ONE. Congratulations! Your manuscript is now with our production department. 

Kind regards, 

on behalf of

Professor Kwasi Torpey 

Academic Editor

PLOS ONE